# Evaluation of VLEs for Binaries of Five Compounds Involved in the Production Processes of Cyclohexanone

**Adriel Sosa [1], Juan Ortega [1,*], Luis Fernández [1], Arturo Romero [2], Aurora Santos [2,*] and David Lorenzo [2]**

[1] Thermal Engineering & Instrumentation Division (IDeTIC), Universidad de Las Palmas de Gran Canaria, 35017 Las Palmas de Gran Canaria, Spain; adriel.sosa@ulpgc.es (A.S.); luis.fernandez@ulpgc.es (L.F.)

[2] Chemical Engineering and Materials Department, Universidad Complutense de Madrid, 28040 Madrid, Spain; aromeros@ucm.es (A.R.); dlorenzo@ucm.es (D.L.)

**\*** Correspondence: juan.ortega@ulpgc.es (J.O.); aursan@ucm.es (A.S.)

**Abstract:** In an attempt to evaluate the separation of certain impurities that arise in some stages of the production of cyclohexanone, this work analyzed the possibility of removing five of these substances via rectification. Due to the scarcity of experimental vapor–liquid equilibrium data for most of the solutions in the effluent of the global process, prior knowledge of their behavior is required. In this work, two predictive models, UNIFAC and COSMO-RS, were used to determine a priori the possibility of obtaining, by distillation, the individual components of seven of the binaries formed by the combination of these five compounds. Since both procedures described quasi-ideal behavior for all the chosen solutions, the results are considered as an approximation, owing to the special nature of the studied systems. The results and characteristics of each system are discussed separately.

**Keywords:** cyclohexanone; vapor–liquid equilibrium; UNIFAC; COSMO-RS; distillation

## 1. Introduction

Annually, around 5000 MT of ε-caprolactam, a monomer of nylon-6, is produced worldwide [1]. Owing to the industrial importance of nylon, especially in the textile industry, different routes have been developed for producing ε-caprolactam in an attempt to reduce the production costs. One of the most commonly used production routes, summarized in Figure 1, consists of converting cyclohexanone into its oxime using a reaction with hydroxylamine or via amoximation (see Figure 1), which is then transformed into caprolactam via Beckmann transposition. This monomer is used to manufacture nylon-6, the fibers of which depend on the impurities present in the caprolactam product; these impurities can arise from the method by which the cyclohexanone is produced, forming in different stages leading to caprolactam and being able to contaminate the final product.

**Figure 1.** ε-caprolactam produced from cyclohexanone via the oxime by a reaction with hydroxylamine and Beckmann transposition.

For example, the oxidation of cyclohexane in the liquid phase generates a product mainly composed of cyclohexanol and cyclohexanone and numerous impurities, including fatty acids, esters, ethers, aldehydes and ketones, which are both linear and cyclic and bear variable numbers of carbon atoms [2].

Figure 2 indicates the different stages in the production of cyclohexanone using cyclohexane as a raw material. These are grouped into three parts, or sections, each involving different chemical engineering processes, which are briefly described.

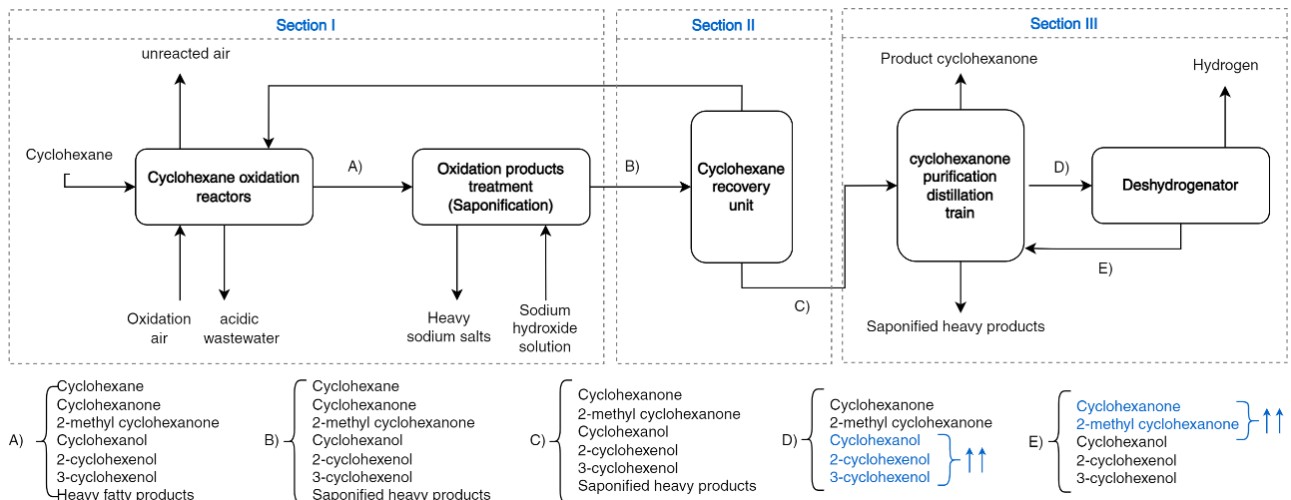

**Figure 2.** Simplified diagram representing the production stages of cyclohexanone, indicating some of the compounds participating in each stream. In blue, the five compounds whose separation is proposed in this work are presented.

Section I: The oxidation of cyclohexane is carried out in a gas–liquid reactor using cobalt salt as a catalyst. Subsequently, the oxidation products are treated in a saponification unit to remove fatty acids and other soluble compounds in the aqueous phase.

Section II: The unreacted cyclohexane, stream B, is recovered in a set of operations and returned to the oxidation unit.

Section III includes the purification train of cyclohexanone and a cyclohexanol dehydrogenation reaction. In this section, cyclohexanol is dehydrogenated to produce cyclohexanone, where the light and heavy impurities produced by oxidation or in the dehydrogenation stage are removed.

The flow-diagram of the process includes the majority of compounds to be separated, cyclohexanone (1) and cyclohexanol (2), as well as some of the derivatives: 2-methylcyclohexanone (3), 2-cyclohexenol (4) and 3-cyclohexenol (5). These latter impurities can react in the oximation and transposition stages, affecting the quality of the caprolactam. Here, we discuss how these impurities could affect the purification of cyclohexanone and the possibility of dragging them out of the product stream resulting from sections II and III. It is necessary to describe the vapor–liquid equilibria (VLE) between the compounds involved, as this will affect the unitary operations in these sections of the process. Hence, three of the ten possible binaries that can be formed were considered less relevant and were excluded from the VLE study. These correspond to 2-cyclohexenol+3-cyclohexenol, due to the chemical similarity of both compounds that gives rise to ideal solutions with similar boiling points, and also the binaries of 2-methylcyclohexanone with the two cyclohexenols mentioned.

The impurities accompanying cyclohexanone do not form part of any industrial process; however, it is interesting to know how they can affect the quality of the caprolactam produced. The scarcity of experimental information for these compounds limits the achievement of our goals. In fact, the only experimental data for cyclohexanone+cyclohexanol and cyclohexanone + 2-methylcyclohexanone systems were measured under vacuum conditions [3]. For this reason, predictive models for the properties involved in the simulation/design of the fractionation operation are required. In this work, Joback's method [4] was employed to estimate the thermophysical properties of the above-mentioned compounds, while the VLE behavior of the corresponding binaries was also investigated and

evaluated at atmospheric pressure. For the latter, the Group Contribution Method (GCM) and UNIFAC (UNIQUAC Functional-group Activity Coefficients), Dortmund's version [5] were used, which are commonly employed for preliminary calculations in chemical engineering and are basic tools in simulation processes with Aspen-Plus© [6]. In addition, the estimates with UNIFAC were compared with those determined using the COSMO-RS (Conductor-like Screening Model for Real Solvents) [7,8], which circumvented the limitation of the GCMs to distinguish between positional isomers, as occurred with the cyclohexenols derived.

The results of the VLE estimations for the binaries form the basis of the discussion on possible strategies to extract the impurities found in cyclohexanone.

## 2. VLE, an Analysis Tool for Separation Processes

An adequate selection of the purification method was based on a previous characterization of the phase equilibria of the studied systems. In this case, knowledge of VLEs was essential to design the separation process of the impurities generated during the production of cyclohexanone. It is not always easy to separate one or more of the compounds in homologous families (as in this case), as these constitute quasi-ideal solutions, and distillation is often the only option available to achieve the desired separation, despite the difficulties presented by some solutions. Before the practice, preliminary information should be obtained using predictive methods on the suitability, or not, of carrying out the experimentation and the possible problems that may arise.

For this, we briefly describe the thermodynamic basis of the models used to predict the VLE of some of the binaries of the main process studied here. In the field of chemical engineering, the set of canonical variables that describes a thermodynamic system, n-tupla $(p,T,\boldsymbol{n})$, is important, and should be calculated. The chemical potential $\mu_i$ $(p,T,\boldsymbol{n})$ provides a criterion for the equilibrium of several phases $\alpha$, $\beta$, ... $\zeta$, depending on the identity.

$$\mu_i^{\alpha}(p, T, \boldsymbol{n}) = \mu_i^{\beta}(p, T, \boldsymbol{n}) = \ldots = \mu_i^{\zeta}(p, T, \boldsymbol{n}) \tag{1}$$

It is known that such an equation can also be written using fugacities. For this case, that of VLE at low or moderate pressures $(0.1 \leq p < 1)$ MPa, the problem is addressed by the *gamma-phi* approach, so called because the non-ideal liquid phase is formulated in relation to the activity coefficient $(\gamma_i)$ and the non-ideality of the vapor by the partial fugacity coefficient, $\hat{\phi}_i$.

$$x_i \gamma_i f_i = y_i \hat{\phi}_i p \tag{2a}$$

$$x_i \gamma_i p_i^o \underbrace{\exp\left[\frac{B_i p_i^o}{RT}\right] \exp\left[\frac{1}{RT}\int_{p_i^o}^{p} v_i^L dp\right]}_{f_i} = y_i p \underbrace{\exp\left[\frac{p}{RT}\left(B_{ii} + \frac{1}{2}\sum_j \sum_k y_j y_k (2\delta_{ji} - \delta_{jk})\right)\right]}_{\hat{\phi}_i} \tag{2b}$$

In Equation (2b), $B_{ii}$ is the second virial coefficient of the i-th compound, which is calculated using generalized correlations or equations of state; $\delta_{mn} \equiv 2B_{mn} - B_{mm} - B_{nn}$, and indicates the non-ideality of the solution formed in the vapor phase; $f_i$ is the fugacity of the i-th compound in the liquid phase, and $p_i^o$ is its vapor pressure. To solve this equation, several properties must be known in advance, such as the critical constants, the molar volumes, $v_i^L$, and the $p_i^o$ of the pure compounds, among others, which were calculated in this work, as shown in Appendix A using the GCMs of Joback [4,9] and Tochigi [10]. Moreover, to solve Equation (2), a suitable model for the activity coefficients must be defined. This matter is addressed in the next section.

### 2.1. Methods to Estimate VLE

For the estimation of properties, especially VLE, a mathematical–thermodynamic tool must be developed that allows a numerical assignment to some properties using previously tested models. Knowledge of the physicochemical properties of some com-

pounds/solutions is important for numerous situations, from basic research to complex calculations, such as the process simulation, and even for the design of equipment. In process engineering, a chemical engineer uses some of the tools developed in the field of thermodynamics, such as GCMs, or UNIFAC/ASOG (Analytical Solutions of Groups), which estimate phase equilibrium properties, among others. They can even predict the VLEs of ternary or multicomponent systems from information generated with the binaries of the species involved. These methods relate the properties with the structural features of the compounds, giving acceptable results for the case of simple molecules. However, other more sophisticated procedures using computational tools [11] or by other means, such as molecular thermodynamics, consider microscopic aspects of the matter.

### 2.2. Activity Coefficient Models

The scarcity of real data of the systems of interest requires the obtainment of preliminary information of their VLEs. Two models were used to estimate the activity coefficients, UNIFAC and COSMO-RS, applied to solve Equation (2). The version of UNIFA-DM by Gmehling et al. [5] is an approximation to the local composition theory, where the activity coefficients are formulated as:

$$\ln \gamma_i = \ln \gamma_i^c + \ln \gamma_i^r \tag{3}$$

Summing two contributions: the combinatorial and the residual, see Appendix B. The former results from the surface and volume interactions among the species that make up the solution and are independent of temperature. This term does not contribute to the calculation of the $p$, $T$-derived quantities, such as the excess enthalpy ($h^E$) or the heat capacity ($c_p^E$). The residual contribution is due to energy interactions, shown in Figure 3a, and is temperature-dependent, yielding the greatest contribution to the original formulation of the model.

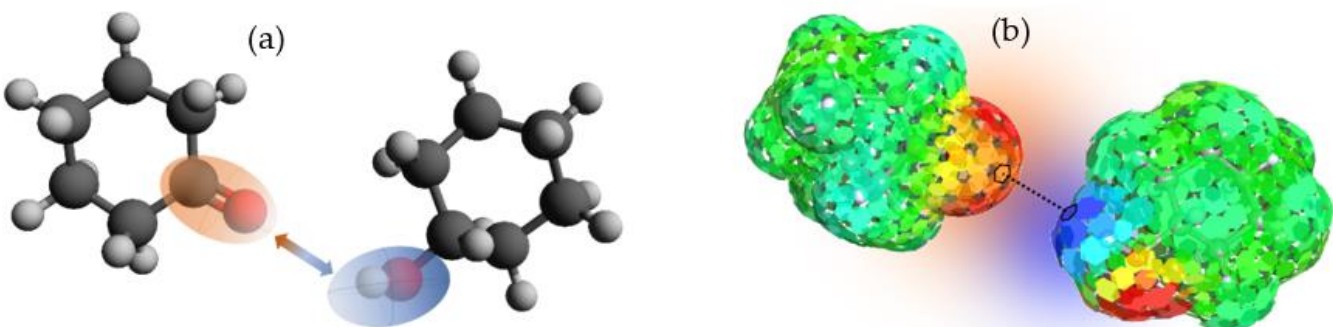

**Figure 3.** (**a**) Interaction between functional groups that give rise to the mixing effects following the GCM, UNIFAC. (**b**) Interaction of hexagonal surface elements that generate $\mu_S$ by means of the COSMO-RS model.

As already mentioned, there are some limitations in the predictions made with the UNIFAC model for the products involved, especially in relation to isomeric compounds. Hence, taking into account the chemical similarity of some of the compounds studied here, the quantum-chemical model COSMO-RS was also used [7,8] for comparison purposes. This model requires the COSMO solvation model to be solved [7] for each species in solution, obtaining the polarization charge density ($\sigma$) on the molecular surface. In a second phase of calculation, statistical thermodynamic methods are used to evaluate the chemical potential for each pair of elements on the molecular surface ($\mu_S$) (see Figure 3b), which have a specific polarization charge density; that is, the potentials of each species in the solution. This is shown in Equation (4), where $p_i(\sigma)$ is the function of the polarization charge density of species i, and $\mu_i^c$ is the combinatorial contribution to the chemical potential [12].

$$\mu_i(T, \mathbf{x}) = \int_\Omega p_i(\sigma)\mu_S(\sigma)\mathrm{d}\sigma + \mu_i^c(T, \mathbf{x}) + RT \ln x_i \tag{4}$$

From this equation, the activity coefficients can be calculated using the following thermodynamic equation:

$$\gamma_{\mathrm{i}} = \exp\left[\frac{\mu_{\mathrm{i}}(T,\mathbf{x}) - \mu_{\mathrm{i}}^{id}(T,\mathbf{x})}{RT}\right] \tag{5}$$

## 3. Results and Discussion

As stated in the introduction, seven binary systems formed by the components constituting the effluent of the cyclohexanone production process are considered. These are: (i) cyclohexanone + 2-methylcyclohexanone, (ii) cyclohexanone + cyclohexanol, (iii) cyclohexanone + 2-cyclohexen-1-ol, (iv) cyclohexanone + 3-cyclohexen-1-ol, (v) 2-methylcyclohexanone + cyclohexanol, (vi) cyclohexanol + 2-cyclohexen-1-ol, and (vii) cyclohexanol + 3-cyclohexen-1-ol.

To assess the efficacy of the theories mentioned in Section 2, VLE data from the literature for systems (i) and (ii) measured under vacuum conditions are used for a preliminary comparison.

### 3.1. Analysis of Experimental VLE Measured under Vacuum Conditions

Figures 4 and 5 show the equilibria for the cyclohexanone + 2-methylcyclohexanone and cyclohexanone + cyclohexanol systems measured at 4.0 and 26.7 kPa, respectively [3]. The first shows ideal behavior and a relative volatility, $\alpha_{\mathrm{ij}} = y_{\mathrm{i}}x_{\mathrm{j}}/y_{\mathrm{j}}x_{\mathrm{i}}$, very close to unity, as deduced from the expression $y_{\mathrm{i}} - x_{\mathrm{i}}$. The UNIFAC-DM and COSMO-RS models also display ideal behavior. However, they generate a systematic shift in the *T-x,y* diagram, probably due to a problem with the vapor pressures measured. In the representation of activity coefficients for this system at $p = 4.0$ kPa (see Figure 4b), it is interesting to observe the high values obtained, which differ from those estimated by both theoretical models (with values of $\gamma_{\mathrm{i}} \approx 1$). A similar observation applies to the comparison of excess Gibbs function, $g^{\mathrm{E}}$. Analogies would apply to the VLE at 26.7 kPa, with estimates of $\gamma_{\mathrm{i}}$ close to unity, shown in Figure 4d. In this case, the predictions and experimental data agree, with the latter being almost randomly distributed around values close to unity. In summary, both models produce similar estimates, both qualitatively and quantitatively.

UNIFAC adequately represents the VLE of the cyclohexanone + cyclohexanol system, shown in Figure 5, although it exhibits higher values of the activity coefficient than for the previous ketone/ketone system, and it also has quasi-ideal behavior. The effect of the pressure on the relative volatility of the binary is significant. Comparing plots of *y-x* vs. *x* at both pressures, the most significant difference at 4.0 kPa is twice that of the curve at 26.7 kPa. In conclusion, it can be established that both models are used to estimate and study the VLE at atmospheric pressure, but, in addition, COSMO-RS differentiates the systems that involve the two isomers of cyclohexenol. UNIFAC predicts the ketone/alcohol system somewhat better, although this result cannot be extrapolated to the remaining five studied systems.

However, if these systems have been chosen as a reference for the verification of theoretical methods, before tackling that goal, we ask ourselves whether the experimentation of the VLEs used is, or is not, correct. The authors [3] argue that the data corresponding to the two systems mentioned in this section are thermodynamically valid, as the consistency is positive with two known methods, that of Frendenslund et al. [13], and that of Van Ness [14]. However, here, we followed the methodology proposed in a previous work [15], globally assessing the quality of the information displayed. In addition to the aforementioned methods, a rigorous procedure [16] was applied, which raised some doubts about the quality of the VLE data, especially due to the instabilities in the equilibrium temperature. Only the binary cyclohexanone + cyclohexanol at 26.7 kPa seems to obey the thermodynamic requirements. Despite this, the scarcity of data in the literature does not offer other option to choose any of the other systems proposed in the study.

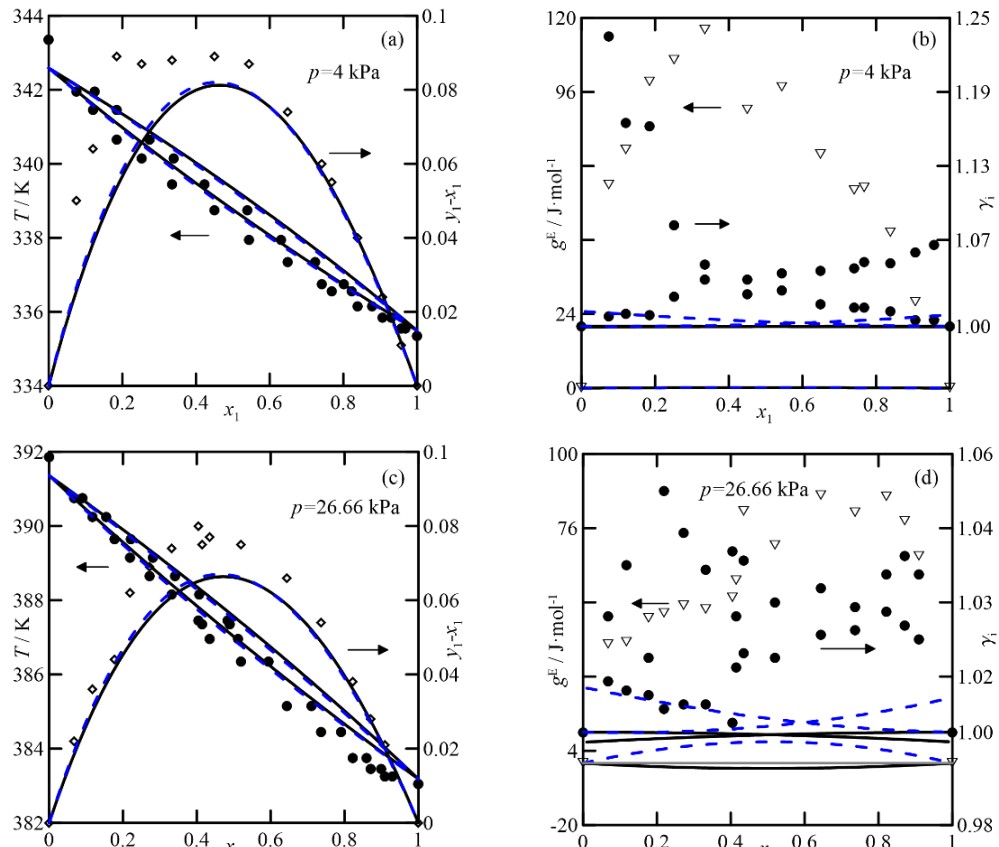

**Figure 4.** Iso-*p* VLE at 4 kPa (**a,b**) and at 26.7 kPa (**c,d**) of cyclohexanone + 2-methylcyclohexanone [3]; (——) UNIFAC-DM; (- - -) COSMO-RS. (●) *T*,x,y (a,c)/$\gamma_i$ (**b,d**); ($^-$) (*y*−*x*), x; (σ) $g^E$/*RT*.

## 3.2. VLE Estimations at p = 101.32 kPa

As mentioned, there are no data in the current literature for systems (i)–(vii), measured at pressures close to atmospheric. Hence, the methods indicated in Section 2 were applied to determine the approximate behavior of those set followed by an individual discussion of each one.

(i) *Cyclohexanone + 2-methylcyclohexanone*: The estimation of the VLE with both theoretical methods shows ideal behavior, as shown in the experiments under vacuum conditions. Nonetheless, the increase in pressure estimates a decrease in the relative volatility of both compounds, as can be seen in the graphs of *T-x,y* and (*y*−*x*) in Figure 6a,b, following the tendency observed at pressures lower than atmospheric.

(ii) *Cyclohexanone + cyclohexanol*: For this binary, the pressure has a more significant influence on the relative volatility of the compounds than in the previous case (see Figure 6a,b). As the purification requirement of both compounds increases, the separation by rectification becomes more complex, requiring many stages. Although the changes in relative volatility with composition, calculated with both models, are similar (see diagram *y*−*x* vs. $x_1$ in Figure 7a), the plot *T-x,y* shows significant differences when relating composition to temperature, with featured effects on the simulation/design process of the operation.

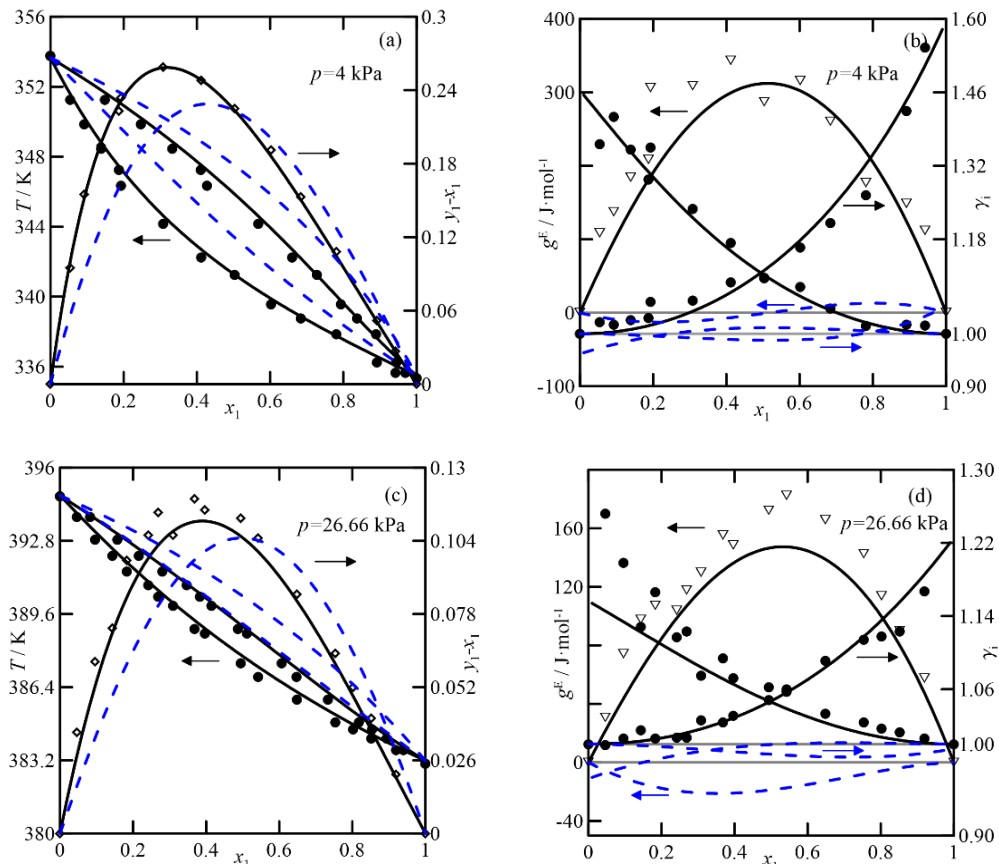

**Figure 5.** Iso-*p* VLE at 4 kPa (**a**,**b**) and 26.66 kPa (**c**,**d**) of the cyclohexanone + cyclohexanol system [3]; (——) UNIFAC-DM; (- - -) COSMO-RS. (●) *T*,x,y (a,c)/$\gamma_i$ (**b**,**d**); (‾) (*y*−*x*), *x*; (σ) $g^E/RT$.

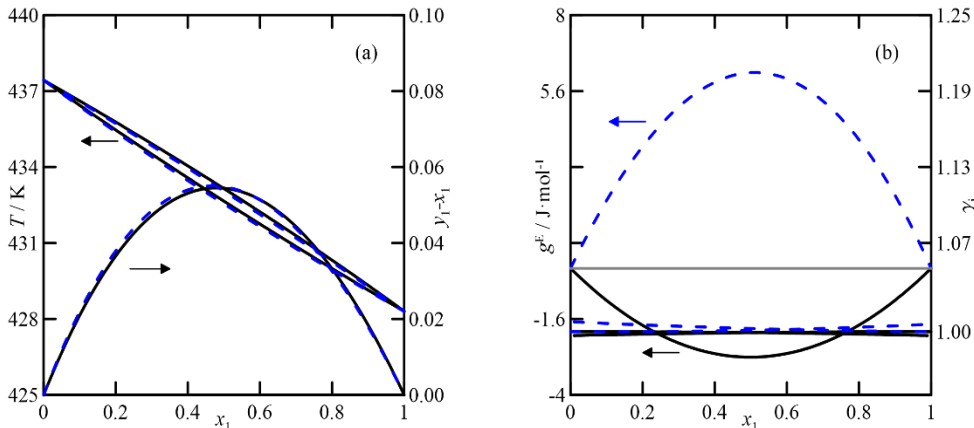

**Figure 6.** Estimations of VLE at 101.32 kPa of cyclohexanone + 2-methylcyclohexanone. (——) UNIFAC-DM; (- - -) COSMO-RS. (**a**) *T*,x,y and (*y*−*x*), *x* (**b**) $\gamma_i$ vs *x* and $g^E/RT$ vs *x*.

(iii/iv) *Cyclohexanone + (2 or 3)-cyclohexen-1-ol:* The VLEs of cyclohexanone with two cyclohexenol isomers are discussed together. The UNIFAC estimation shows ideal behavior for both systems (see Figure 8), and those corresponding to the two isomers are identical, as UNIFAC does not distinguish between them. However, the results generated by COSMO-RS are different, as the model predicts a certain non-ideality in the liquid phase, showing a negative deviation of the Raoult´s law and a light inflexion in the alcohol-rich zone, according to the diagrams *T-x*,y and ($y_1 - x_1$) − *x*, shown in Figure 8c. This theory slightly differentiates the behavior of the binary with each alcohol isomer. In summary, the

estimation indicates that the rectification of this mixture, like the previous ones, is complex due to the low relative volatility of its components and even more so due to the presence of an azeotrope predicted using the quantum-chemical method.

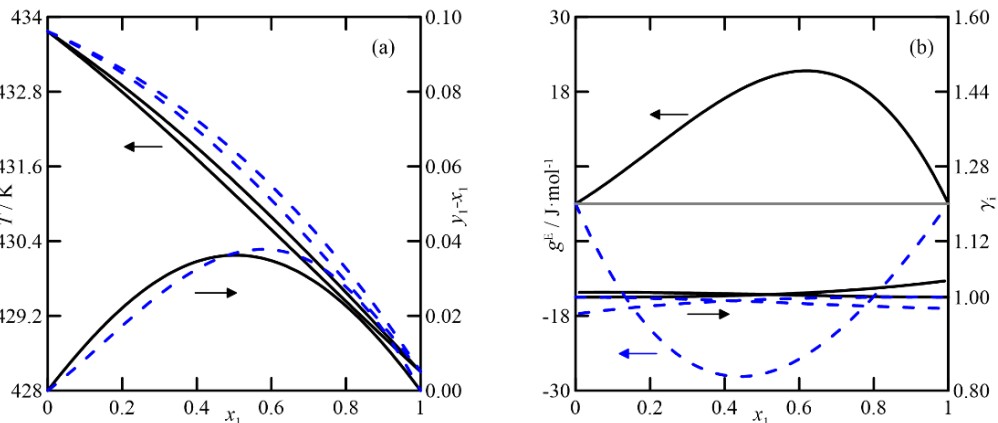

**Figure 7.** Estimations of VLE at 101.32 kPa of cyclohexanone + cyclohexanol. (—) UNIFAC-DM; (- - -) COSMO-RS. (**a**) $T$,x,y and $(y-x)$, $x$ (**b**) $\gamma_i$ vs $x$ and $g^E/RT$ vs $x$.

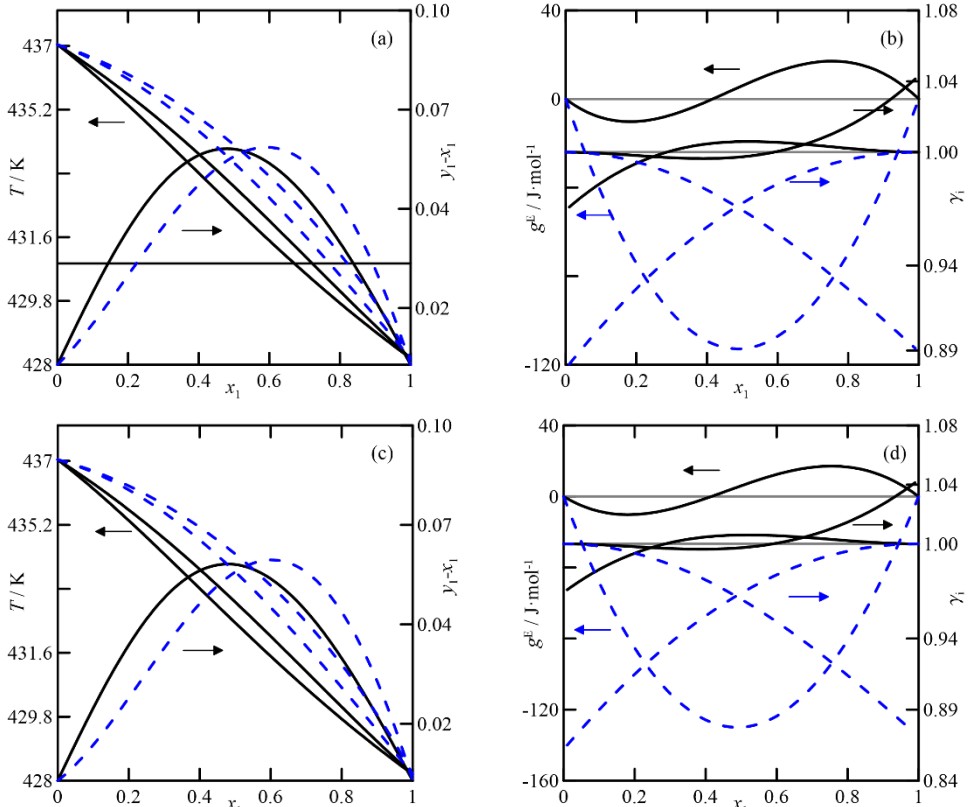

**Figure 8.** Estimations of VLE at 101.32 kPa of: (**a**,**b**), cyclohexanone + 2-cyclohexen-1-ol, and (**c**,**d**), cyclohexanone + 3-cyclohexen-1-ol. (—) UNIFAC-DM; (- - -) COSMO-RS.

It is clear that the chosen model determines the design of the separation equipment. That is, if the simulation is performed with the COSMO-RS model, the composition of cyclohexanone in the residue stream will not be negligible due to the low relative volatility of both components in the zone corresponding to $x_1 \to 0$. Moreover, this fact will also imply the need for numerous stages in the stripping section of the column.

(v) *2-methylcyclohexanone + cyclohexanol*: The COSMO-RS estimation indicates that the liquid phase behaves ideally. In addition, the fact that the boiling points of the pure compounds differ by more than 3 °C makes the differences between the compositions of the liquid and vapor phases close to zero. The UNIFAC model also predicts the ideal behavior of the liquid phase, but with lower activity coefficients than those estimated using COSMO-RS. These results provide VLE behavior of this system with a fold in the zone rich in 2-methylcyclohexanone, generating a quasi-azeotrope at atmospheric pressure, as shown in Figure 9a,b.

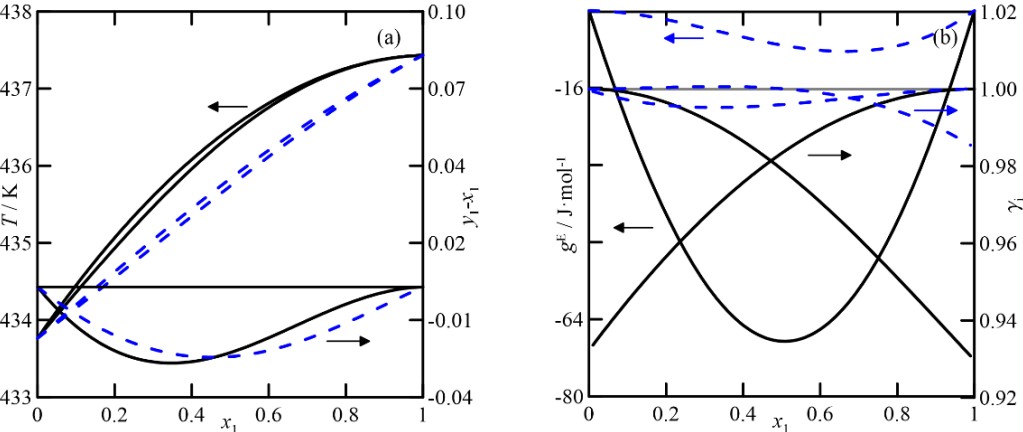

**Figure 9.** Estimations of VLE at 101.32 kPa of 2-methylcyclohexanone + cyclohexanol, (—) UNIFAC-DM; (- - -) COSMO-RS. (**a**) $T$,x,y and $(y-x)$, $x$ (**b**) $\gamma_i$ vs $x$ and $g^E/RT$ vs $x$.

(vi/vii) *cyclohexanol + (2 or 3)-cyclohexen-1-ol*: As expected, the systems composed of pairs of cyclic alcohols constitute ideal solutions, shown in Figure 10, the behavior of which is represented by both models. In the 2-methylcyclohexanone + cyclohexanol system, the representation of the activity coefficients with UNIFAC gives rise to a more folded *T-x,y* diagram than that generated by COSMO-RS, reflecting the difficulty of separating cyclohexanol from its derived compounds.

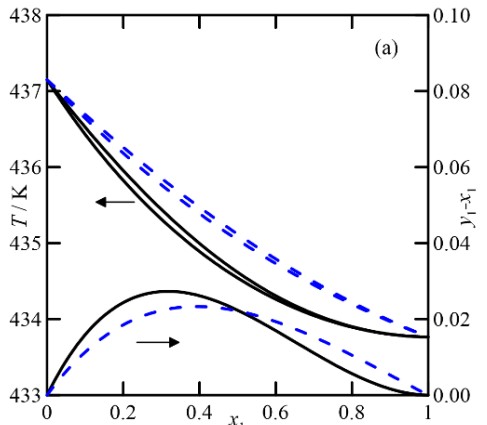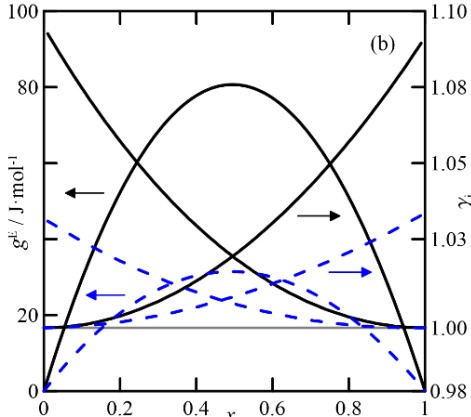

**Figure 10.** *Cont.*

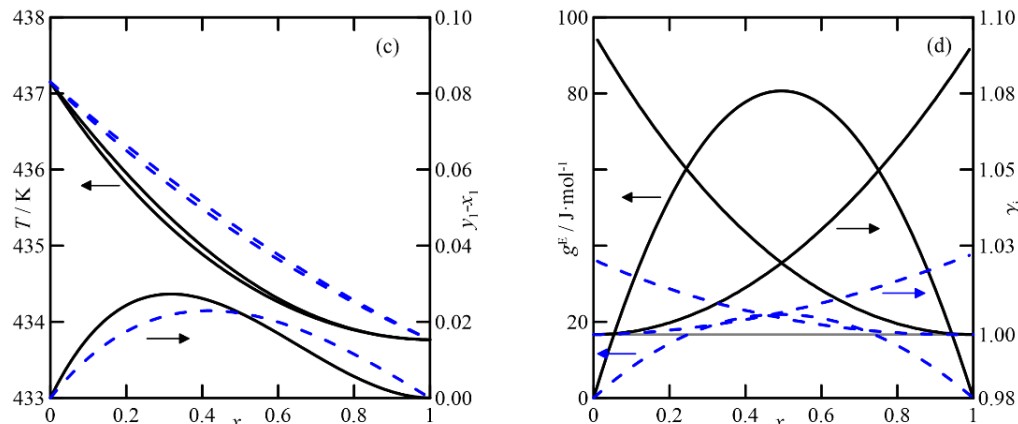

**Figure 10.** VLE at 101.32 kPa of: (**a**,**b**) cyclohexanol + 2-cyclohexen-1-ol, and (**c**,**d**) cyclohexanol + 3-cyclohexen-1-ol. (——) UNIFAC-DM; (- - -) COSMO-RS.

### 3.3. About the Possibilities of Distillation of the Systems Studied from the Results Obtained

Despite particular differences in the predictions of the VLEs using UNIFAC and COSMO-RS, both predict the low relative volatility of the components of the solutions considered, as shown in Figure 11. In all cases, the minimum and maximum values of this parameter are close to the practical guideline limits, between 0.95/1.05 and 0.90/1.1 [17,18], which advise on viable separation via simple rectification. However, the separation of these binaries, or a stream involving a more complex system, requires the use of advanced distillation operations to achieve greater efficiency in the purity of the final products.

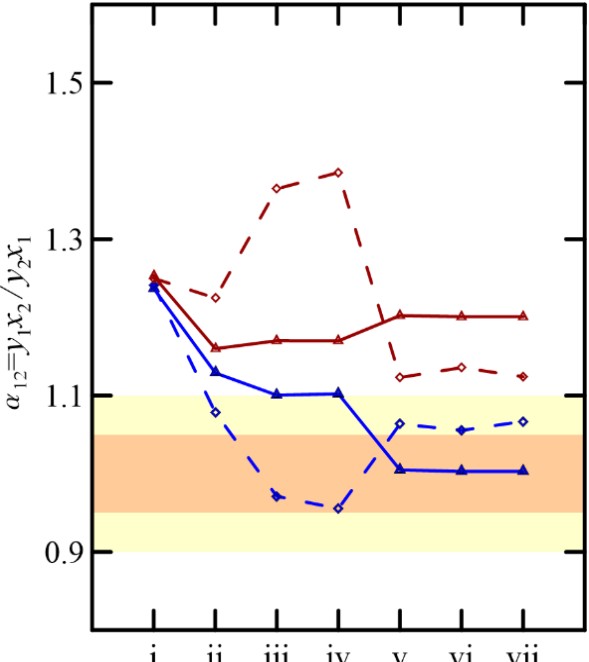

**Figure 11.** Minimum (——) and maximum limits (——) of the relative volatility estimated with UNIFAC (continuous) and COSMO-RS (dashed lines). Distillation not recommended, ▨ complex separation ▨ .

Since the pressure of the system conditions the relative volatility, distillation under vacuum is one of the alternatives to be explored. In azeotropic or quasi-azeotropic systems, a design with two columns operating at different pressures, in a *pressure-swing* operation, should be considered.

As showed previously, the VLEs of the systems studied here present negligible sensitivity to the pressure changes, except for the cyclohexanone + cyclohexanol system. This is also supported by the relationship between vapor pressure and temperature for these compounds. Figure 12 represents that relationship using Antoine's equation with *reduced coordinates*, having estimated the parameters of that equation through Tochigi's GCM [10] (see Appendix A), as the experimental information was not found in the literature. It is observed that the slopes of the representation $\log p^o_{i,r}$ vs. $1/T_r$ are similar for almost all the compounds considered, and this negatively affects the change in relative volatility as the pressure of the system varies, thus limiting the usefulness of this operational alternative.

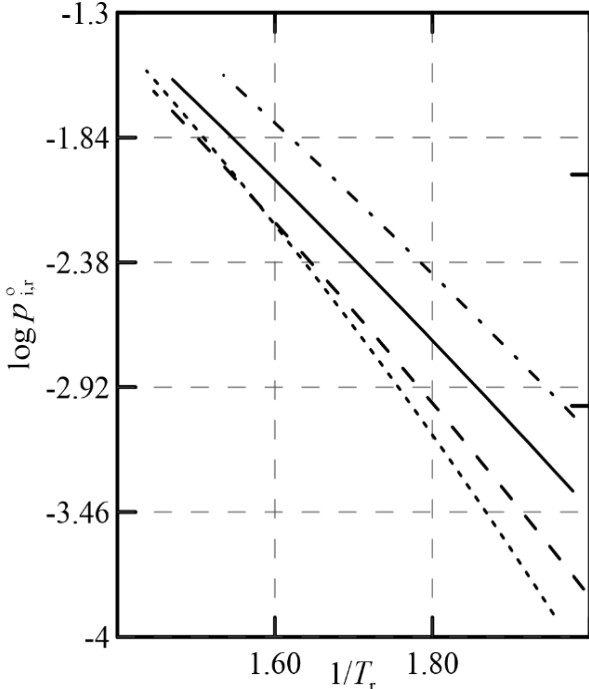

**Figure 12.** Plot of vapor pressure of the compounds: cyclohexanone (− − − −); cyclo- hexanol (● ● ●); 2-methylcyclohexanone (− · −); 2-cyclohexen-1-ol (− − −), in reduced coordinates.

Other options to separate the studied systems are extractive distillation and/or distillation via the saline effect. Both have the same purpose: to use an agent (entrainer) that alters the activity coefficients of the liquid phase in order to increase the relative volatility between the main components of the mixture. In the first case, the agent could be a solvent with a high boiling point, while in the case of the saline effect, it is a soluble salt that remains in a liquid stream throughout the column. However, both operations would increase the cost of the overall process, as when an entrainer is used, at least one additional column is required to purify the head and/or bottom effluents, while the saline effect requires an evaporator or precipitator to remove salt from the bottom stream.

## 4. Conclusions

Because of their presence in some intermediate purification stages, a set of seven binary systems were selected from the global production process of cyclohexanone. Due to the scarcity of experimental VLE data, only two of them have been studied under vacuum [3], so estimates were made for all of them to determine a priori the possibility of obtaining the individual components via distillation. The VLE estimates at atmospheric pressure for the cyclohexanone + 2-methylcyclohexanone and cyclohexanone + cyclohexanol systems obtained with the COSMO-RS and UNIFAC-DM models provide information comparable to the experimental one, reflecting quasi-ideal behavior, that is, with slight deviations in the values of $\gamma = 1$, and folding in the $T$ vs. $x,y$ diagram.

In other cases, such as those of cyclohexanone solutions with cyclohexenol isomers, the two models predict different behaviors; even the quantum-chemical model establishes differences for the two isomers, a fact that UNIFAC does not reflect. However, without the corresponding experimentation, this cannot be confirmed. The discrepancy between both models could be due to their instabilities, although both define the quasi-ideal behavior of the studied solutions.

From a practical perspective, the work carried out to design the rectification operations is only considered as an approximation due to the special and complex nature of the equilibria studied, revealing the difficulty in separating these systems. As shown in Figure 11, the maximum and minimum volatilities are simultaneously over the operation limits suggested for only two systems, namely cyclohexanone + 2-methylcyclohexanone and cyclohexanone + cyclohexanol, for the binaries that offer real information. Our findings suggest a priori that extractive distillation appears to be the most suitable technique compared to pressure-swing; however, the final decision will also depend on economic criteria and other alternatives that could be found in the future.

**Author Contributions:** Conceptualization, J.O., A.S. (Adriel Sosa) and A.R.; methodology, A.S. (Adriel Sosa) and J.O.; software, L.F.; validation, A.S. (Adriel Sosa)., L.F., J.O., A.R., A.S. (Aurora Santos) and D.L.; investigation, A.S. (Adriel Sosa), A.R. and J.O.; resources, J.O.; data curation, A.S. (Adriel Sosa) and L.F.; writing—original draft preparation, A.S. (Adriel Sosa), J.O. and A.R.; writing—review and editing, J.O., L.F., A.R., A.S. (Aurora Santos) and D.L.; supervision, J.O. All authors have read and agreed to the published version of the manuscript.

**Funding:** This research received no external funding.

**Data Availability Statement:** The new data used in this article were calculated according to Appendix A.

**Conflicts of Interest:** The authors declare no conflict of interest.

## Appendix A  Properties Estimation of the Pure Compounds

Joback's method [4] is a reevaluation of Lydersan's GCM [19] in which new groups are incorporated and the contributions of each are reparametrized. That method provides independent equations to predict critical properties ($T_c$, $p_c$, $v_c$) and other thermophysical quantities separately, except for $T_c$, the estimation of which depends on the normal boiling point, either obtained experimentally or estimated using the corresponding equation. The expressions used are as follows:

$$T_b(K) = 198 + \sum_k N_k \tau_{bk} \tag{A1}$$

$$T_c(K) = T_b \left[ 0.584 + 0.965 \left\{ \sum_k N_k \tau_{ck} \right\} - \left\{ \sum_k N_k \tau_{ck} \right\}^2 \right]^{-1} \tag{A2}$$

$$p_c(bar) = \left[ 0.113 + 0.0032 \cdot N_{atoms} - \sum_k N_k \pi_{ck} \right]^{-2} \tag{A3}$$

$$v_c(cm^3 mol^{-1}) = 17.5 + \sum_k N_k v_{ck} \tag{A4}$$

where $N_k$ corresponds to the number of k-type groups in the molecule and $\tau_{bk}$ $\tau_{ck}$, $\pi_{ck}$, $v_{ck}$ and $\Delta h_{ck}$ correspond, respectively, to the normal boiling point, critical temperature, critical pressure, critical volume and vaporization enthalpy. An extract of the group parameters of Joback's method is shown in Table A1 for the compounds studied. In addition to the mentioned properties, the evaluation of the second virial coefficient by Tsonopoulos' correlations [20], required to characterize the non-ideality of the vapor phase, requires the dipolar moments, $\mu$, of the molecules. In the literature, $\mu$-values were found for cyclohexanone and 2-methylcyclohexanone [21,22] and for cyclohexanol [21,23]. The remaining values were estimated using Gaussian 03W software [24] with the computational technique B3LYP, belonging to the DFT (Density Functional Theory) and orbital 6–31 G(d)

family of methods. Table A1 summarizes all the values calculated in this work and the data found in the literature.

**Table A1.** Poperties of the pure compounds studied.

| | $T_c$ (K) | $p_c$ (kPa) | $v_c$ (dm$^3$·mol$^{-1}$) | $Z_c$ | $\mu$ (D) | $a$ | $b$ | $T_b$ (K) |
|---|---|---|---|---|---|---|---|---|
| cyclohexanone | 656.02 [a] | 4379.60 [a] | 0.312 [a] | 0.251 | 3.06 [b,g] | −0.0191 | 0.0000 | 428.72 [a] |
| | 654.00 [c] | 4660.00 [c] | | | 3.08 [c] | | | 428.80 [c] |
| | 653.00 [d] | 4000.00 [d] | | | | | | 428.84 [d] |
| | 629.15 [e] | 3850.00 [e] | 0.311 [e] | 0.229 [e] | 3.08 [e] | | | 428.90 [e] |
| | | | | | 3.05 [f] | | | |
| 2-methyl-cyclohexanone | 670.64 [a] | 3721.50 [a] | 0.367 [a] | 0.245 | 3.07 [b] | −0.0150 | 0.0000 | 446.93 [a] |
| | | | | | 3.00 [f] | | | |
| cyclohexanol | 643.39 [a] | 4521.18 [a] | 0.323 [a] | 0.273 [a] | 1.56 [f] | 0.0878 | 0.0274 | 448.41 [a] |
| | 625.00 [c] | 3700.00 [c] | | | 1.86 [c] | | | 434.35 [c] |
| | 625.15 [e] | 3749.00 [e] | 0.322 [e] | 0.232 [e] | | | | 434.00 [e] |
| | 625.15 [i] | 3749.00 [i] | | | | | | 433.65 [i] |
| 2-cyclohexen-1-ol (‡) | 633.48 [a] | 4533.30 [a] | 0.319 [a] | 0.275 [a] | 1.47 [f] | 0.0878 | 0.0259 | 444.03 |
| | | | | | | | | 437.00 [h] |

[a] Joback [4,9] (Equations (A1)–(A4)); [b] [22]; [c] [25]; [d] [26]; [e] [23]; [f] B3LYP; 6–31 G(d); [g] [21]; [h] [27]; [i] [28]. (‡) in the context of Joback, there is no distinction between positional isomers. These estimations are therefore identical for 3-cyclohexen-1-ol, except for the dipolar moment ($\mu$ = 1.3687 D) and consequently the parameter $b$ of Tsonopoulos ($b$ = 0.0236).

Antoine's constants for 2-cyclohexen-1-ol could not be found in the literature; these were estimated by the GCM of Tochigi et al. [10]. Both the estimated constants and data obtained from the literature are summarized in Table A2. Constants were calculated using the following expressions:

$$A' = \log 101.32 + \frac{B'}{T_b / K + C' - 273.15} \tag{A5a}$$

$$A' = A_0 + \sum_k N_k A_k \tag{A5b}$$

$$B' = B_0 + \sum_k N_k B_k \tag{A6}$$

$$C' = C_0 + \sum_k N_k C_k \tag{A7}$$

The original work by Tochigi et al. recommends using Equation (A5a) to calculate $A'$, reserving Equation (A5b) for when experimental data for the normal boiling point of the compound are not available.

Equations (A1)–(A3) are used to evaluate, together with the parameters of Antoine's equation (Table A2), the acentric factor by means of the equation:

$$\omega = -1 - \log p_r^o|_{T_r=0.7} = -(1+a) + b/(0.7-c) \tag{A8}$$

where $a$, $b$ and $c$ are constants of Antoine's equation in reduced form, deduced by Ortega et al. [29].

$$a = A - \log p_c \quad b = B/T_c \quad c = C/T_c \tag{A9}$$

**Table A2.** Values for the parameters of Antoine's equation to calculate the vapor pressures.

|  | *A* | *B* | *C* | *Range T/K* | *Ω* | Ref. |
|---|---|---|---|---|---|---|
| cyclohexanone | 6.0832 (2.539) | 1477.73 (2.283) | 65.89 (0.1004) | 318.15–428 | 0.256 | [3] |
|  | 6.5950 | 1832.20 | 28.95 | 270–430 |  | [25] |
|  | 6.1066 | 1498.18 | 63.25 | 345–458.34 |  | [26] |
| 2-methyl-cyclohexanone | 6.1092 (2.539) | 1527.67 (2.278) | 65.27 (0.0973) | 338.76–436.94 | 0.256 | [30] |
|  | 6.0736 | 1495.61 | 70.0874 | 318.15–437.45 |  | [3] |
| cyclohexanol | 6.0580 (2.402) | 1261.89 (1.961) | 122.36 (0.1902) |  | 0.444 | |
|  | 6.066 | 1258.75 | 123.67 | 320–435 |  | [31] |
|  | 13.564 * | 2689.90 * | 133.31 |  |  | [26] |
|  | 5.9290 | 1199.10 | 128.15 | 370–440 |  | [25] |
| 2-cyclohexen-1-ol | 6.2100 (2.701) | 1483.70 (2.342) | 84.25 (0.1330) |  | 0.528 | Equations (A7)–(A9) |

\* Constants adapted to Antoine's expression expressed as its natural logarithm. Values between parenthesis indicate Antoine's constants expressed in reduced form, Equation (A9).

**Appendix B  UNIFAC Model**

The terms corresponding to the combinatorial and residual contribution of the UNIFAC model for the activity coefficient, Equation (3), are below. The first term is a function of the molecular volume and surface.

$$\ln \gamma_i^c = \ln \frac{\varphi_i}{x_i} + 1 - \frac{\varphi_i}{x_i} - 5q_i \left( \ln \frac{\Phi_i}{\vartheta_i} + 1 - \frac{\Phi_i}{\vartheta_i} \right) \tag{A10}$$

Surface fractions ($\vartheta_i$) and volume ($\Phi_i$) are normalizations of molecular surfaces ($q_i$) and volumes ($r_i$), respectively.

$$\Phi_i = \frac{x_i \cdot r_i}{\sum_j x_j \cdot r_j}, \; r_i = \sum_k v_k^{(i)} R_k, \; \varphi_i = \frac{x_i \cdot r_i^{3/4}}{\sum_j x_j \cdot r_j^{3/4}}, \; \vartheta_i = \frac{x_i \cdot q_i}{\sum_j x_j \cdot q_j}, \; q_i = \sum_k v_k^{(i)} Q_k \tag{A11}$$

where $Q_m$ y $R_m$ are the group surface and volume, respectively; $v_k^{(i)}$ is the number of groups of type k in the i-th molecule. The parameters of the pure compounds of this work are compiled in Table A3 The residual term is the difference between the group contribution to the activity coefficient in the mixture, $\Gamma_k$, and in the pure compound, $\Gamma_k^{(i)}$. This way, boundary conditions for this property are enforced.

$$\ln \gamma_i^r = \sum_k v_k^{(i)} \left( \ln \Gamma_k - \ln \Gamma_k^{(i)} \right) \tag{A12}$$

Each of the above terms correspond to the energy interactions between molecular surface elements. Hence, these are a function of the group surface fractions in the solution, $\Theta_m$ and the pairwise group interaction energy, $\psi_{mn}$.

$$\ln \Gamma_k = Q_k \left[ 1 - \ln \left( \sum_m \Theta_m \psi_{mn} \right) - \sum_m \frac{\Theta_m \psi_{km}}{\sum_n \Theta_n \psi_{nm}} \right] \tag{A13}$$

$\Theta_m$ is the result of a normalization of the group surfaces by the fraction of each group in the solution, $X_m$.

$$\Theta_m = \frac{X_m Q_m}{\sum\limits_n X_n Q_n}, \quad X_m = \frac{\sum\limits_j \nu_m^j x_j}{\sum\limits_j \sum\limits_n \nu_n^j x_j} \tag{A14}$$

The pairwise group interaction energy is a function of temperature and, according to Gmheling's modification of UNIFAC [5], is given by:

$$\psi_{mn} = \exp\left(-\frac{a_{mn} + b_{mn}T + c_{mn}T^2}{T}\right) \tag{A15}$$

The binary interaction parameters in Equation (A15) are recorded in Table A4 for all group combinations related to the pure components included in this work.

**Table A3.** UNIFAC groups—volume and surface parameters [5].

| | Groups | id | $\nu_k$ | $R_k$ | $Q_k$ |
|---|---|---|---|---|---|
|  | c-CH$_2$ | 42 | 4 | 0.7136 | 0.8635 |
| | CH$_2$C=O | 9 | 1 | 1.7048 | 1.5542 |
|  | c-CH$_2$ | 42 | 4 | 0.7136 | 0.8635 |
| | c-CH | 42 | 1 | 0.3479 | 0.1071 |
| | CH$_3$ | 1 | 1 | 0.6325 | 1.0608 |
| | CH$_2$C=O | 9 | 1 | 1.7048 | 1.5542 |
|  | c-CH$_2$ | 42 | 4 | 0.7136 | 0.8635 |
| | c-CH | 42 | 1 | 0.3479 | 0.1071 |
| | OH | 5 | 1 | 1.0630 | 0.8663 |
|  | c-CH$_2$ | 42 | 4 | 0.7136 | 0.8635 |
| | c-CH | 42 | 1 | 0.3479 | 0.1071 |
| | CH=CH | 2 | 1 | 1.2832 | 1.2489 |
| | OH | 5 | 1 | 1.0630 | 0.8663 |

**Table A4.** UNIFAC group binary interaction parameters [5].

| Group id | Group id | $a_{m,n}$ | $b_{m,n}$ | $c_{m,n}$ | $a_{n,m}$ | $b_{n,m}$ | $c_{n,m}$ |
|---|---|---|---|---|---|---|---|
| 1 | 2 | 189.66 | −0.2723 | 0.0 | −95.418 | 0.6171 | 0.0 |
| 1 | 5 | 2777.0 | −4.6740 | 0.001551 | 1606.0 | −4.7460 | 0.0009181 |
| 1 | 9 | 433.60 | 0.1473 | 0.0 | 1987.0 | −4.6150 | 0.0 |
| 1 | 42 | −680.95 | 4.0194 | −0.006878 | 1020.8 | −6.0746 | 0.01015 |
| 2 | 5 | 2649.0 | −65.5080 | 0.004822 | 1566.0 | −5.8090 | 0.005197 |
| 2 | 9 | 179.80 | 0.6991 | 0.0 | 92.811 | −0.7171 | 0.0 |
| 2 | 42 | −78.190 | 0.1327 | 0.0 | 182.40 | −0.3030 | 0.0 |
| 5 | 9 | −250.00 | 2.8570 | −0.006022 | 653.30 | −1.4120 | 0.0009540 |
| 5 | 42 | 3856.0 | −17.970 | 0.02083 | 3246.0 | −4.9370 | −0.001143 |
| 9 | 42 | 156.53 | −0.7135 | 0.0 | 498.92 | −0.0440 | 0.0 |

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
