# Peer review of "Evaluation of VLEs for Binaries of Five Compounds Involved in the Production Processes of Cyclohexanone"

_2305-7084, doi:10.3390/chemengineering6030042_

Round 1

Reviewer 1 Report

Reviewer’s report

In this manuscript, the authors attempt to evaluate the separation of several impurities that appear during the production of cyclohexanone and their further removal by means of distillation process. Because of the lack of available experimental liquid-vapor equilibrium data, the authors apply UNIFAC and COSMO-RS predictive models/tools. Prediction results are presented for a number of binary systems containing cyclohexanone and cyclohexanol, as well as some of the derivatives: 2-methylcyclohexanone, 2-cyclohexenol and 3-cyclohexenol. Thermodynamic properties and phase diagrams for the binary systems under study are discussed.

The manuscript can be accepted for publication after minor revisions. The following are my comments:

  1. Note to the captions to Figures 4 and 5. Explain the symbols (experimental data) clearly.
  2. Check up the color (blue dashed lines) for COSMO-RS in captions for Figures 7-10.
  3. Figure 11 is duplicated, there is no caption for Figure 12.
  4. Line 131 – Give the abbreviation ASOG definition.
  5. Line 394 “Both the estimated constants and data obtained from the literature are summarized in Table A2”. It should be clear for readers, please specify the values in parentheses. The columns in Table A2 should be formatted.
  6. Please add new Appendix for UNIFAC model. Insert tables with UNIFAC group assignment in this study (main groups, subgroups and their stoichiometry). Please add the matrix with UNIFAC group interaction parameters used in this work. It is important for readers because the geometrical and energy parameters for all groups under consideration were taken from the literature source [Ref.5]. In this case, refer to the Appendix in the text body.

Reviewer 2 Report

In the article presented for review, the authors described their research very meticulously and with great care. All of the results contained in it were presented in a very clear and legible way for the recipient. The conclusions and the study summary are consistent and well-formulated. Summing up, the reviewed work presents a very high substantive and experimental value.

Comments for authors:

The errors that should be corrected are only cosmetic errors that do not affect the quality of the reviewed work in any way:

1 / the designation of equation 2b should be on the same line as the equation, the font of the equation should be slightly reduced

2 / spaces should be removed next to (a) and (b) – Figure 3

3/  line 216 the word claim in red - it should be changed to black

                 „(…)The authors [3] claim that the data corresponding to the (…)”

4/ the drawings on pages 11 and 12 should be given the next number, Figures 12 and 13 and a reference in the text

5/ old literature should be partially replaced with new ones
